# Sensitivity Characterization of Cascaded Long-Period Gratings Operating near the Phase-Matching Turning Point

**DOI:** 10.3390/s20215978

**Published:** 2020-10-22

**Authors:** Wei Zhou, Yanli Ran, Zhijun Yan, Qizhen Sun, Chen Liu, Deming Liu

**Affiliations:** 1Institute of Microscale Optoelectronics (IMO), Shenzhen University, Shenzhen 518060, China; zhou_wei@hust.edu.cn; 2School of Optical and Electronic Information, National Engineering Laboratory for Next Generation Internet Access System, Huazhong University of Science and Technology, Wuhan 430074, China; yanzhijun@hust.edu.cn (Z.Y.); qzsun@hust.edu.cn (Q.S.); liuchen@mail.hust.edu.cn (C.L.); dmliu@hust.edu.cn (D.L.)

**Keywords:** long period fiber grating, optical fiber sensors, phase-matching turning point, quality factor

## Abstract

We characterized a cascaded long-period gratings (LPGs)-based sensor that was operating at the phase-matching turning point (PMTP). The cascaded LPGs constructed an in-fiber Mach–Zehnder interferometer (MZI), which exhibited a series of high-quality-factor (*Q*) narrow-bandwidth resonance peaks. As the LPG operated at the PMTP, the proposed sensor showed an ultrahigh refractive index (RI) and temperature sensitivity, and high measurement precision. In this study, we took an in-depth look at the effects of grating separation on Q-factor and sensitivity. The results showed that the sensitivity to the surrounding refractive index (SRI) reached 4741.5 nm/RIU at 1.4255 and 2138 nm/RIU, over the range of 1.335–1.373. In addition, the temperature sensitivity was around 4.84 nm/°C. With a 0.02 nm wavelength resolution, the RI and temperature sensing limits were 9.3 × 10^−6^ RIU and 5.5 × 10^−3^ °C.

## 1. Introduction

Single-molecule detection and ultrasensitive sensing are critically important for understanding biochemical materials and processes. Optical sensing is a direct, fast, low-cost, and highly sensitive detection technique that has attracted significant interests for the development of high-sensitivity and high-accuracy sensors. Several ultrasensitive optical sensors were developed, such as polarization interferometer sensors [1], tunable diode laser sensors [2], and optical fiber sensors [3]. Of these, the optical fiber-based sensors received the most attention because of their advantages of simple structure, facile fabrication, and high sensitivity and accuracy.

There are several types of optical fiber sensors, including fiber-grating-based sensors [4], optical fiber interferometers [5], and distributed fiber sensors [6], and these sensors are applied in biochemical sensing [7], structure-health monitoring [8], seismic detection [9], and so on. With grating sensors, long-period gratings (LPGs) have the capacity to couple the fundamental core mode to the forward-propagating cladding mode, which results in a series of attenuation peaks in the transmission spectrum (mostly used for high-sensitivity biochemical measurements). The attenuation peaks of LPGs are sensitive to the surrounding perturbations [10], and their sensitivity to refractive index (RI) is mode-dependent, which leads to a low sensitivity for most biochemical solutions (a typical RI is around 1.34 RIU).

Several methods were proposed to improve the aqueous-solution RI sensitivity of LPGs, including coating the gratings with a higher RI material to increase the equivalent surrounding RI [11], reducing the cladding size so that the effective cladding-mode index is closer to the sensing solution [12,13], and designing the gratings so that the cladding mode operates near the phase-matching turning point (PMTP) [14]. Of these techniques, the PMTP method is the simplest way to enhance the RI sensitivity [10]. However, the attenuation peak of the LPG at the PMTP has a very broad bandwidth and a poor Q factor, which degrade the sensing accuracy.

Although LPG-based Mach–Zehnder interferometers (MZIs) have multiple resonant peaks within the profile of a single LPG loss peak, which could greatly improve the LPG Q factor [15], their sensitivity remains very low. In 2018, Feng theoretically proposed an extra-high sensitive fiber sensor based on cascaded LPGs operating near the PMTP [16]. In this work, we experimentally fabricate and investigate the transmission and sensing features of a cascaded LPG-based sensor operating near the PMTP. Benefit from the LPG working at near the PMTP, the proposed sensors showed a high sensitivity to both the surrounding refractive index (SRI) and temperature, and the cascaded structure could improve the Q factor of sensor.

## 2. Fabrication and Experimental Characteristics of the Cascaded LPG-Based Sensor

This section describes the sensing principle, fabrication method, and spectral characterization of the cascaded LPG-based sensors.

### 2.1. Working Principle

LPG could couple the fiber core mode into the forward-propagating cladding mode, and in turn, it could also couple the cladding mode back to the core mode. Such reversible coupling behavior makes the LPG to construct an MZI. In this work, we experimentally fabricate a cascaded LPGs-based MZI sensor. Figure 1 shows the configuration of a cascaded LPG-based sensor, which includes two identical LPGs with 3 dB coupling efficiency, separated by a distance *D*.

Here, Λ is the period of the cascaded LPGs, and *L* is the length of a single LPG. The periodic modulation of LPGs excites coupling between the core and the cladding mode. In Figure 1, the first LPG couples 50% light energy of the core mode into the cladding mode, which is recoupled back into the fiber core by the second LPG, thereby forming the two arms of MZI, due to the different light paths in the cladding and the core. The intensity *I* of the interference signal was [17,18,19].
(1)$$I=I_{1}+I_{2}+2\sqrt{I_{1}I_{2}}\cos\left(\Phi^{m}\right)$$
(2)$$\Phi^{m}=\frac{2\pi \Delta n_{\mathrm{eff}}^{m}L_{\mathrm{eff}}}{\lambda }$$
where $I_{1}$ and $I_{2}$ are the intensities of the light propagating along the fiber core and cladding, respectively, $\Phi^{m}$ is the phase difference between the core and the cladding modes, $\Delta n_{\mathrm{eff}}^{m}=n_{\mathrm{eff}}^{\mathrm{core}}-n_{\mathrm{eff}}^{\mathrm{clad},m}$ is the difference between the effective RI of the fundamental core and that of the $m^{\mathrm{th}}$ cladding mode, where $n_{\mathrm{eff}}^{\mathrm{core}}$ and $n_{\mathrm{eff}}^{\mathrm{clad},m}$ are the effective refractive indices of the fiber core and fiber cladding, respectively, λ is the central wavelength of the input light, and $L_{\mathrm{eff}}$ is the effective cavity L + D. According to the MZI theory, the peak wavelength λ of constructive interference is [20]
(3)$$\lambda =\frac{2\Delta n_{\mathrm{eff}}^{m}L_{\mathrm{eff}}}{2m+1}$$

The free spectral range of the LPG-based MZIs is [21]
(4)$$s=\frac{\lambda^{2}}{L_{\mathrm{eff}}\Delta n_{\mathrm{eff}}}$$

According to Equation (3), the wavelength shift Δλ of the cascaded LPG-based sensor is [22]
(5)$$\lambda =\frac{1}{1-\left(\frac{dn_{\mathrm{eff}}^{\mathrm{core}}}{d\lambda }-\frac{dn_{\mathrm{eff}}^{\mathrm{clad},m}}{d\lambda }\right)\Lambda }\times \left(\frac{\left(\delta n_{\mathrm{eff}}^{\mathrm{core}}-\delta n_{\mathrm{eff}}^{\mathrm{clad},m}\right)}{n_{\mathrm{eff}}^{\mathrm{core}}-n_{\mathrm{eff}}^{\mathrm{clad},m}}+\frac{d\Lambda }{\Lambda }\right)\frac{2\Delta n_{\mathrm{eff}}^{m}L_{\mathrm{eff}}}{2m+1}$$
where the first factor of Equation (5) accounts for waveguide dispersion; the second factor describes the environmental dependence of waveguide dispersion; and the third factor accounts for material expansion caused by the temperature. According to the calculation, the wavelength of turning point of the 21st cladding mode is within the C-band. By employing Equation (5), we simulated the temperature and RI sensitivity of the 21st cladding mode versus the wavelength around the turning point (see Figure 2a,b). As shown in Figure 2, both the temperature and RI sensitivity increased exponentially, as the resonance wavelength was close to the turning point, which was verified in the experiment.

### 2.2. Fabrication and Characterization of Cascaded LPG-Based Sensor

In this work, the cascaded LPG-based sensor was UV inscribed on a PS1250/1500 fiber, which is a photosensitive single-mode fiber, by using the point-by-point inscription (see Figure 3), in which a shutter and motorized stage were used to control the LPG period, and a 248 nm excimer laser operated at a high repetition rate was treated as a quasi-continuous-wave laser. A broadband visible-infrared light source and an optical spectrum analyzer (OSA) were used to measure the transmission spectrum of the LPG-based sensor (Figure 4).

### 2.3. Q Factor of Cascaded LPGs

The *Q* factor is one of the most important parameters of an optical fiber sensor because it is the accurate indicator of the measurement resolution [23]. The *Q* factor of the cascaded LPG pair was given as [23,24,25].
(6)$$Q=\frac{2L_{\mathrm{eff}}\Delta n_{\mathrm{eff}}}{\lambda }$$
where λ is the wavelength, and $\Delta n_{\mathrm{eff}}$ is the effective RI difference between the fundamental core mode, and the cladding mode.

In the experiment, we fabricated four cascaded LPGs with grating separations of 2 cm, 3 cm, 4 cm, and 5 cm. The transmission spectrum shows that the bandwidth of the resonance peaks of the cascaded LPGs decreased linearly with increasing grating separation, and the *Q* factor increased simultaneously, in which the Q factors of the cascaded LPGs with 2 cm, 3 cm, 4 cm, and 5 cm separations were 35.8, 41.6, 53.3, 19.4, respectively. Compared with a conventional dual-peak LPG [26], the bandwidth and *Q* factor of the cascaded LPGs with 5 cm separation were improved about four times. Although further increasing the separation gap should significantly increase the *Q* factor, the larger separation would increase the transmission loss of the cladding mode and reduce the interference peaks, thereby, affecting the performance of the sensor.

## 3. Temperature and RI Sensing Characteristics and Discussion

This section presents the characterization of the sensor performance, in detail in terms of temperature and RI sensitivity of the cascaded LPGs operated at the PMTP.

### 3.1. Temperature Sensitivity

The ambient temperature typically is an important parameter for the operation of an optical fiber sensor, so the detection of this parameter is imperative for the development of numerous applications [27,28,29]. LPGs have a relatively high temperature sensitivity, especially the LPGs operating at the PMTP. In this experiment, we investigated cascaded LPGs-based sensors from 20 °C to 60 °C, using a thermoelectric controller, as shown in Figure 5.

In the experiment, three cascaded LPs sensors with different separation (2 cm, 3 cm, and 4 cm) were tested. We monitored the resonance peaks shift under different temperatures. The results are shown in Figure 6a–f. The experimental results showed that all resonance peaks at the long-wavelength (short-wavelength) side of the turning point showed redshift (blueshift). Table 1 lists the resonance wavelength and the temperature sensitivity of each peak.

The simulation results in Figure 2 indicate that the sensitivity increased as the resonance peak approached the PMTP. In addition, the experimental results showed that the resonance peaks close to the PMTP had higher temperature sensitivity, compared to those farther from the PMTP (see Table 1). Each of these three samples had a grating period about 154 μm, and the PMTP near 1570 nm; however, they had different grating separations, which only affect the resonance peak position and *Q* factor. Peaks 2 and 3 of the 2-cm-separated LPG pair were first-order symmetrical resonance peaks at the PMTP. The LPG pairs with 3 and 4 cm separations exhibited symmetrical resonance peaks 1 and 2. A much higher temperature sensitivity might be achieved by monitoring the wavelength separation of the two symmetrical resonance peaks. The temperature sensitivity of the LPG pairs with 2, 3, and 4 cm grating separations was 4.84 nm/°C (peaks 2 and 3), 2.32 nm/°C (peaks1 and 2), and 2.361 nm/°C (peaks1 and 2), respectively. We plotted all experimental data together, see Figure 7, which could further verify that the resonance peak closer to the PMPT had the higher sensitivity.

### 3.2. Refractive Index Sensitivity

The experimental setup shown in Figure 8 was used to characterize the RI of the cascaded LPG-based sensor. The setup consisted of a supercontinuum light source, a thermoelectric controller (SLD70-SN9DNA with 0.001 °C temperature control accuracy) to control the sensor temperature, and an OSA. During the experiment, both ends of the fiber were held straight so as to be free of strain, and different RI solutions (containing water and glycerin) were used in the sensing area.

The high temperature sensitivity of the proposed sensor would affect the RI measuring accuracy. To eliminate the temperature crosstalk, during the RI sensing process, the cascaded LPG sensors was placed inside incubator with 0.001 °C temperature control accuracy, which was better than the temperature measuring accuracy of the proposed sensor. Once the external RI of a cascaded LPG-based sensor changed, the spectrum of the sensor shifted due to the change in effective RI of the cladding modes. Figure 9 plots the transmission spectrum of the cascaded LPG-based sensors with different SRIs. With increasing SRI, the peak to the left (right) of the PMTP showed blueshifts (redshifts). We individually analyzed different interference peaks from the LPG-based sensor with different RIs, and the transmission spectra were plotted in Figure 9a–d, for grating separations of 2, 3, 4, and 5 cm, respectively. As shown in Figure 9, the SRI affected the position of the resonance peaks, and the amplitude of the resonance peaks changed irregularly with increasing SRI. Given the limited wavelength range of the light source, the maximum RI detected was 1.4255.

Figure 10 shows wavelength shift of interference peaks of cascaded LPGs with different grating separations versus RI value from 1.335 to 1.4255. Two sensitivity regions appeared—a linear sensing region (1.335–1.373) and a nonlinear sensing region (1.373−1.4255). However, in most biosensing applications, the effective RI range was 1.335−1.373. For the interference peaks on each side of the PMTP, the wavelength shift increased as the peak wavelength approached 1570 nm, so the sensitivity increased as the wavelength peak approached the PMTP. Of the four samples, only the sample with the 3 cm separation between LPGs had a resonance peak located near the PMTP (around 1583 nm in air), so the above analysis indicated that this cascaded LPG pair would have the higher RI sensitivity.

With increasing RI, the peak at the PMTP split into two peaks (peak 1 and peak 2), which had an RI sensitivity of −1067 and 1071 nm/RIU, respectively, in the linear RI-sensing region (1.335–1.373). By monitoring the wavelength separation of peaks 1 and 2, the RI sensitivity might be doubled to around 2138 nm/RIU in the RI range 1.335–1.373 and 4741.5nm/RIU at RI = 1.4255.

The RI sensitivity of interference peaks of the proposed sensors with various grating separations was calculated; shown in Table 2. The results clearly showed that the peaks closer to the PMTP had a greater sensitivity. According to Equation (5), the sensor with the larger grating separation should have a greater RI sensitivity. In the experimental results, peaks 4, 3, 2, and 3 for grating separations of 2, 3, 4, and 5 cm, respectively, had almost the same resonance wavelength of 1654 nm (the underlined data in Table 2), and the RI sensitivities were 582, 755, 816, and 843 nm/RIU, respectively, in the linear RI sensing region (1.335–1.373), which was consistent with Equation (5).

We also compared the sensing performance of the proposed cascaded LPGs with other types of LPG sensors. The cascaded LPG sensors operated near the PMTP had the highest sensitivity, with both the RI and temperature sensitivity enhanced over tenfold. The experiment results also indicated that the longer cavity length of the cascaded LPG sensors had a higher sensitivity and better Q-factor to improve the sensing accuracy. Furthermore, by monitoring the wavelength separation of two interference peaks, the highest sensitivity was about 3683.4 nm/RIU in the RI range 1.335–1.4255. Based on the 0.02 nm wavelength resolution of OSA, the minimum detectable change in RI was about 9.35 × 10^−6^ RIU, which was better than a commercial refractometer. Such a high RI sensitivity would promote the cascaded LPG-based sensor suited for highly sensitive biochemical sensing applications.

## 4. Conclusions

In this work, we experimentally demonstrated a high-sensitivity precision RI and temperature sensor based on two cascaded identical LPGs operated at the PMTP. Both the simulated and experimental results showed that the resonance peak of the cascaded LPGs near the PMTP had the highest temperature and RI sensitivity of all resonance peaks. As the interference peaks on each side of the PMTP shifted in opposite directions (in terms of wavelength), their separation might be monitored in the proposed sensor to achieve an average sensitivity of the surrounding RI up to 2138 nm/RIU in the linear sensing RI region 1.335–1.373, and the maximum sensitivity of the surrounding RI reached 4741.5 nm/RIU at RI = 1.4255 in the nonlinear sensing region. The sensor also provided a temperature sensitivity of 4.84 nm/°C in the range 20–60 °C. Moreover, the experimental results showed that the temperature sensitivity did not depend on the grating separation and, inversely, the RI sensitivity depended on grating separation (a greater grating separation provided greater RI sensitivity). Based on spectral interrogation, the minimum detectable change in RI and temperature was 9.3 × 10^−6^ RIU and 5.5 × 10^−3^ °C, respectively, at 0.02 nm wavelength resolution. This high sensitivity and measurement precision of the cascaded LPG-based sensors should allow them to be used for biochemical sensing.

## Figures and Tables

**Figure 1 sensors-20-05978-f001:**
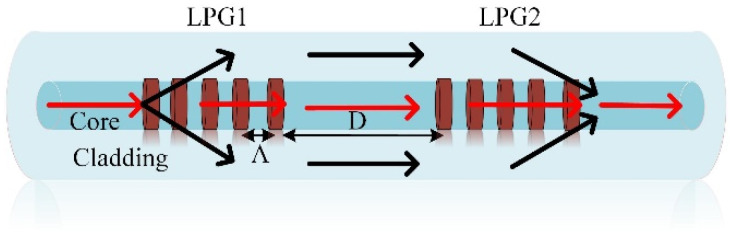
Configuration of cascaded LPG-based sensor.

**Figure 2 sensors-20-05978-f002:**
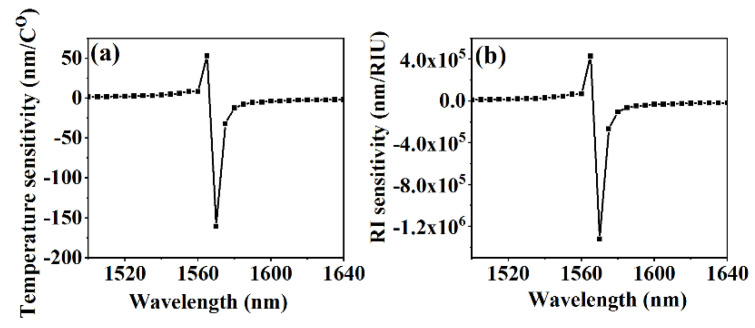
Simulation results of (**a**) temperature and (**b**) RI sensitivity of 21st cladding mode.

**Figure 3 sensors-20-05978-f003:**
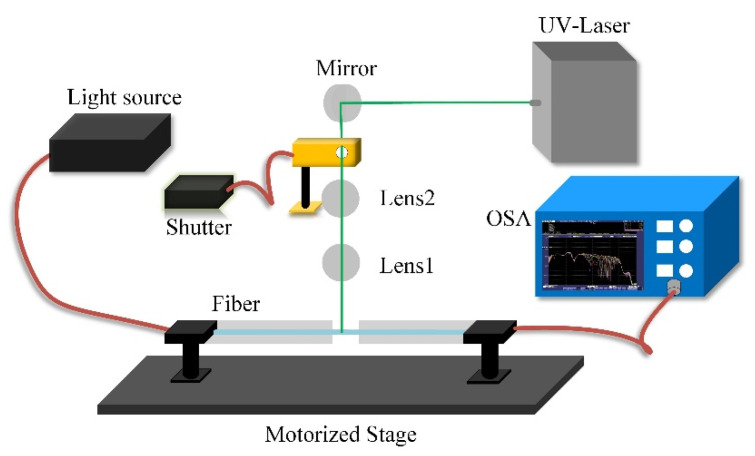
Fabrication process for cascaded LPG-based sensor.

**Figure 4 sensors-20-05978-f004:**
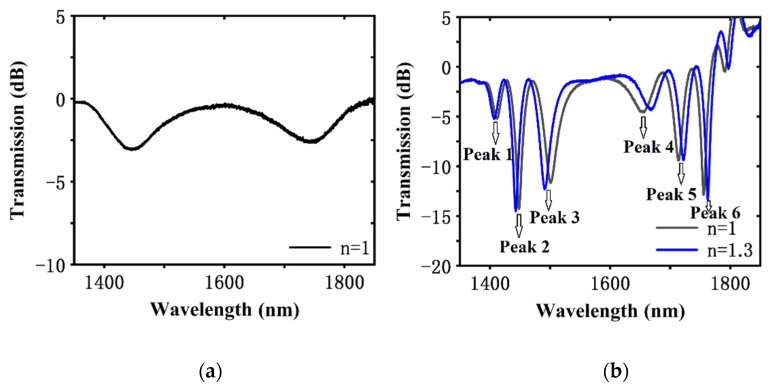
Spectrum of (**a**) LPG1 and (**b**) LPG pair operating at the PMTP.

**Figure 5 sensors-20-05978-f005:**
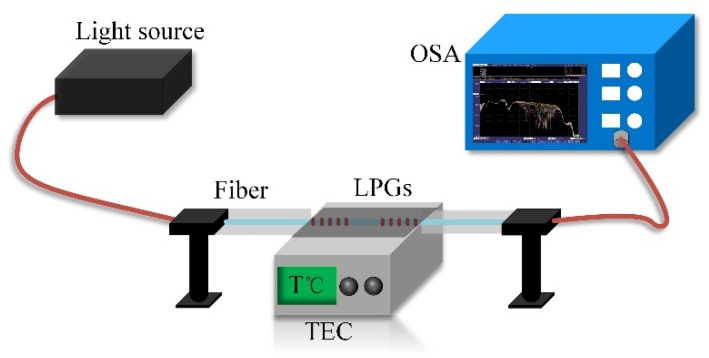
Temperature-sensing setup “TEC” is a thermoelectric controller.

**Figure 6 sensors-20-05978-f006:**
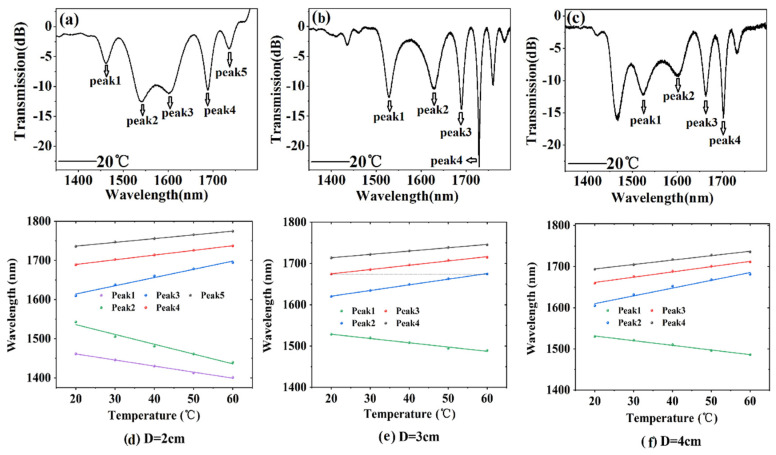
Variation of peak wavelength and spectrum vs. temperature of the cascaded LPG-based sensors with different separations (**a**,**d**) 2 cm, (**b**,**e**) 3 cm, (**c**,**f**) 4 cm.

**Figure 7 sensors-20-05978-f007:**
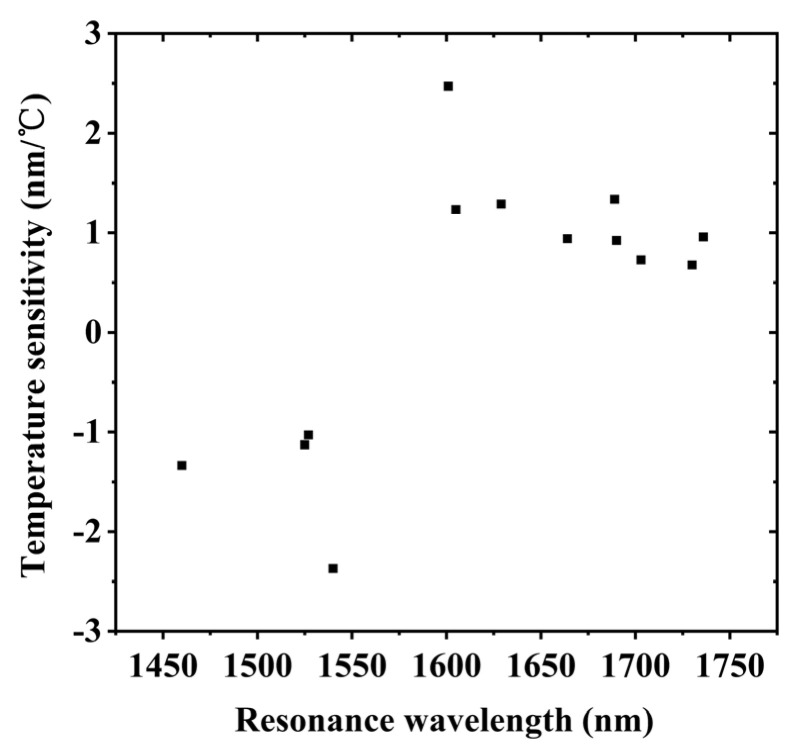
Temperature sensitivity as a function of resonance wavelength.

**Figure 8 sensors-20-05978-f008:**
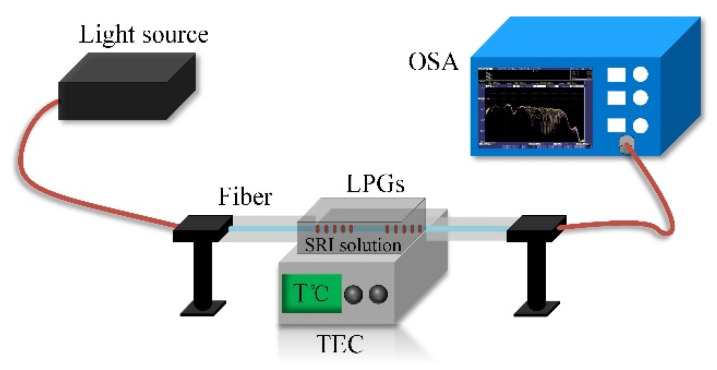
SRI sensing initial setup. “TEC” is a thermoelectric controller.

**Figure 9 sensors-20-05978-f009:**
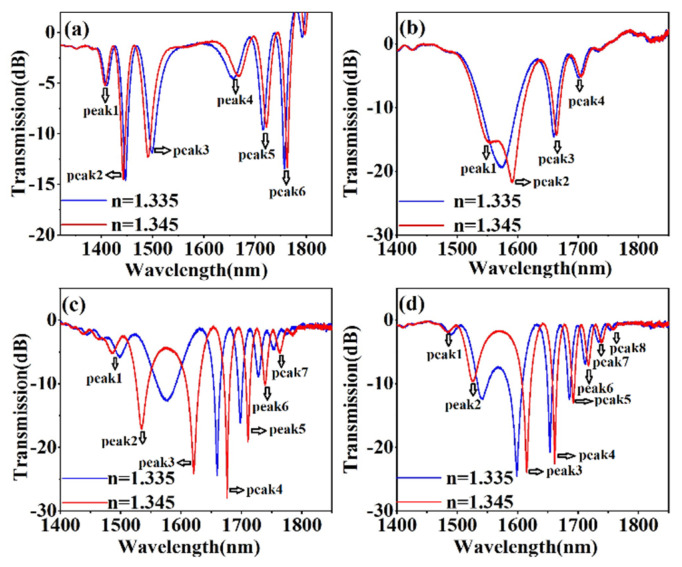
SRI responses Δλ of the cascaded LPGs-based sensors with different separation gaps—(**a**) 2 cm, (**b**) 3 cm, (**c**) 4 cm, and (**d**) 5 cm.

**Figure 10 sensors-20-05978-f010:**
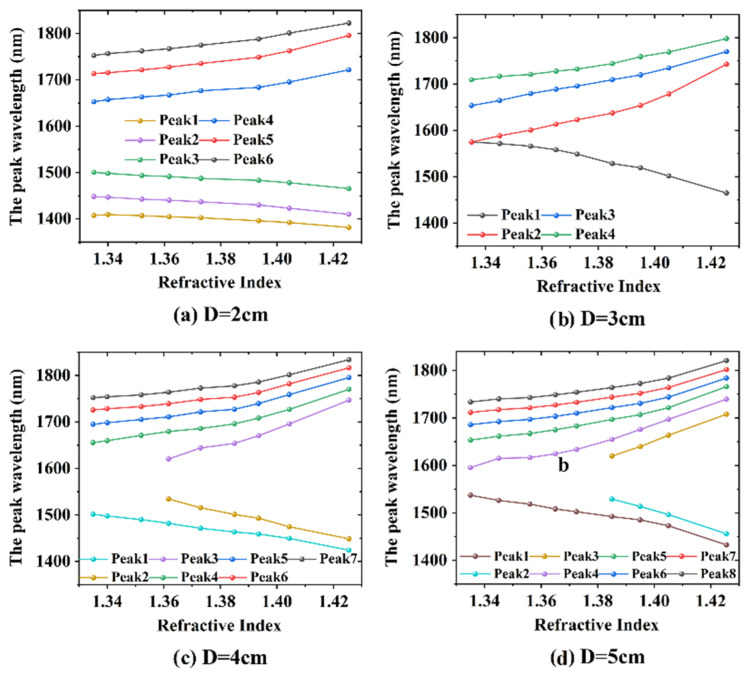
Wavelength shift of interference peaks versus SRI for cascaded LPGs with grating separations of (**a**) 2 cm, (**b**) 3 cm, (**c**) 4 cm, and (**d**) 5 cm.

**Table 1 sensors-20-05978-t001:** Temperature-sensing parameters for each peak.

Peak	D = 2 cm		D = 3 cm		D = 4 cm	
	λ_res_ (nm)	Ts (nm/°C)	λ_res_ (nm)	Ts (nm/°C)	λ_res_ (nm)	Ts (nm/°C)
1	1460	−1.335	1527	−1.03	1525	−1.128
2	1540	−2.37	1629	1.29	1605	1.233
3	1601	2.47	1690	0.923	1664	0.94
4	1689	1.338	1730	0.678	1703	0.73
5	1736	0.958				

**Table 2 sensors-20-05978-t002:** RI-sensing parameters for each peak.

Peak	D = 2 cm		D = 3 cm		D = 4 cm		D = 5 cm	
	λ_res_ (nm)	R_s_ (nm/RIU)	λ_res_ (nm)	R_s_ (nm/RIU)	λ_res_ (nm)	R_s_ (nm/RIU)	λ_res_ (nm)	R_s_ (nm/RIU)
1	1407	−227	1572	−1067	1504	−782	1538	−814
2	1449	−420	1572	1071	1654	816	1596	897
3	1498	-435	1654	755	1694	676	1654	843
4	1654	582	1771	602	1725	575	1685	650
5	1713	527			1759	524	1711	547
6	1754	483					1735	518

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
