# Peer review of "Sensitivity Characterization of Cascaded Long-Period Gratings Operating near the Phase-Matching Turning Point"

_sensors, 2020, doi:10.3390/s20215978_

Round 1
Reviewer 1 Report
This manuscript shows the sensitivity characterization for a cascaded long-period gratings operating around the phase-matching turning point. My comments are following:
- There are several redundant contents in this manuscript such as "Material and methods should be described in sufficient detail to allow..." (page 2), "we fabricated four such cascaded LPGs with grating separation of 2, 3, 4, and 5 cm" (repeat twice in page 4). Authors should carefully check and revise this manuscript.
- For the manuscript simplification, I think that Table 1 and Table 4 are not necessary to be highlighted.
- Authors should describe why the temperature sensitivity is negative in peak 1 and peak 2 for 2 cm separation and peak 1 for 3 cm separation and peak 1 for 4 cm separation. What is the operating mechanism?
- There are several typing and grammar errors such as "The cascaded LPG pair was constructed in a fiber Mach-Zehnder interferometer and exhibits a series..." (in abstract), "The experiment tested three cascaded LPGs sensors (separations of 2, 3, and 4), Figure 6 (a)...", and "Figure 10 plot the peak...", etc. Therefore, I think this manuscript requires to be revised for publishing in Sensors.
Author Response
Reviewer 1
This manuscript shows the sensitivity characterization for a cascaded long-period gratings operating around the phase-matching turning point. My comments are following:
(1). There are several redundant contents in this manuscript such as "Material and methods should be described in sufficient detail to allow..." (page 2), "we fabricated four such cascaded LPGs with grating separation of 2, 3, 4, and 5 cm" (repeat twice in page 4). Authors should carefully check and revise this manuscript.
> We are very sorry for these careless mistakes. We have checked the manuscript again and revised all mistakes. “Material and methods should be described in sufficient detail to allow...” (page 2) and the sentence “we fabricated four such cascaded LPGs with grating separation of 2, 3, 4, and 5 cm” in line 128 in page 4 were deleted. And we added “In the experiment, we have fabricated four cascaded LPGs with grating separations of 2cm, 3cm, 4cm, and 5 cm.” in line 138-139. All revised parts have been highlighted by using “Tracking Changes” function.
(2). For the manuscript simplification, I think that Table 1 and Table 4 are not necessary to be highlighted.
> Thanks the reviewer’s suggestion. We have deleted the 2 tables and add the description in the context. We added “in which the Q factors of cascaded LPGs with 2cm, 3cm, 4cm, and 5 cm separations are 35.8, 41.6, 53.3, 19.4, respectively.” in line 141-143 which is highlighted by using “Tracking Changes” function.
(3). Authors should describe why the temperature sensitivity is negative in peak 1 and peak 2 for 2 cm separation and peak 1 for 3 cm separation and peak 1 for 4 cm separation. What is the operating mechanism?
> That is very good question. In the manuscript, we have given the numerical description of sensitivity of cascaded LPG.
As shown in Formula (5), the first term of equation is the waveguide dispersion, which is usually called as γ factor. It is an important parameter to analyze the sensing property of LPG. For LPG, there is a turning point in the waveguide dispersion curve. Before the turning point, γ factor is positive, and after the turning point, γ factor is negative. The second term is environment dependent wavelength dispersion which are always negative value as changing of RI and temperature. The third term is the wavelength information, which is always positive.
The dual-peak of LPG is the cladding resonance peak at the turning point, which has the same cladding mode order, but peak at the short wavelength is before the turning point and has positive γ factor, and peak at the longer wavelength is after the turning point and has negative γ factor. That is why the peak 1 and peak 2 for 2 cm separation and peak 1 for 3 cm separation and peak 1 for 4 cm separation have a negative temperature sensitivity, which all are at the shorter wavelength side. So the RI and temperature sensitivity is negative, when the cladding mode order is before the turning point. After the turning point, the sensitivity is positive.
(4). There are several typing and grammar errors such as "The cascaded LPG pair was constructed in a fiber Mach-Zehnder interferometer and exhibits a series..." (in abstract), "The experiment tested three cascaded LPGs sensors (separations of 2, 3, and 4), Figure 6 (a)...", and "Figure 10 plot the peak...", etc. Therefore, I think this manuscript requires to be revised for publishing in Sensors.
> Thank you very much for your suggestions. We have carefully checked and revised the manuscript. All revised parts have been highlighted by using “Tracking Changes” function.
“The cascaded LPG pair was constructed in a fiber Mach-Zehnder interferometer and exhibits a series...” (in abstract) had been revised to “The cascaded LPGs construct an in fiber Mach–Zehnder interferometer (MZI), which…”.
“The experiment tested three cascaded LPGs sensors (separations of 2, 3, and 4), Figure 6 (a)...” had been revised to “In the experiment, cascaded LPs sensors with three different separation (2cm, 3cm and 4cm) were tested. We have monitored the resonance peaks shift under different temperature. The results are shown in Fig.6 (a)-(f).” in line 159-160.
“Figure 10 plot the peak...” had been revised to “Figure 10 shows wavelength shift of interference peaks of cascaded LPGs with different grating separations versus RI value from 1.335 to 1.4255." in line 211-212.

Reviewer 2 Report
The authors experimentally demonstrated a high sensitivity Mach-Zehnder interferometer based on two cascaded LPG operating at the PMPT. The working principle is based on a theoretically paper by Feng et al. published in 2018, which is ref. 16 of the manuscript. This research topic might be potentially interesting to the sensor or telecom community. However, the authors should clear some points before the manuscript can be accepted. Below are some questions and comments from the referee.
- It seems this manuscript is based on a published conference paper in 2019 18th International Conference on Optical Communications and Networks (ICOCN), Highly sensitive RI and temperature sensor based on cascaded LPGs at near the phase-matching turning point. The authors should be clear about what is new in the present manuscript.
- This work is based on the theoretical design of ref. 16 of the manuscript. However, in the experimental results section, the authors did not compare their results to the theoretical prediction of ref. 16. Can the authors make a comparison? What is the optimum sensitivity of this design? Why did the authors choose this wavelength range for the experimental demonstration? What is the optimum design?
- p.2, line 52. “To verify this theory, we experimentally investigate in this work the transmission and sensing features of a cascaded LPG-based sensor operating near the PMTP.”
The description is not clear, although the referee could get the point the author would like to say.
4. p.2, line 66. “As presented in [16], the resonance peak of LPG operated at PMTP would have an ultra-high sensitivity, due to the waveguide dispersion trend to the infinite.”
Please be more specific. What does “trend to the infinite” mean?
- p2, line 60-62. This paragraph is not needed.
- 6.For the simulation result of Fig.2, why use 21st cladding mode, is there any specific reason?
- P.3, line99. What is a PS1250/1500 fiber?
- P.6, line 165. “These results also show that the bandwidth of the resonance peak of the cascaded LPG pair narrows with increasing temperature”
How to see this from Fig. 7? How to see the bandwidth change? There is no changing temperature. Only temp sensitivity.
From the results of Fig.7, it seems the temp sensitivity is about the same in the range between 1600 and 1750 nm. The authors should relate this to their findings above.
9. From the results in Table 4, it seems the sensitivity is highest when grating separation is 3 cm. It is not changing monotonically as the authors described in the conclusion (p.9, line 233). Can the author comment on why it’s highest at 3 cm? How many times did the authors repeat the measurement?

Author Response
Point 1: It seems this manuscript is based on a published conference paper in 2019 18th International Conference on Optical Communications and Networks (ICOCN), Highly sensitive RI and temperature sensor based on cascaded LPGs at near the phase-matching turning point. The authors should be clear about what is new in the present manuscript.
Response 1: Thank you very much for your suggestion. Yes, this manuscript is based on the conference paper in ICOCN 2019. In the conference paper, we only demonstrated the simple experiment results, which expressed our latest research quickly. But it didn’t include the theoretical analysis and fabrication, even the detailed discussion of experiment results. I believe that this manuscript is total new manuscript. There is not any conflict of interest between these two manuscripts.
Point 2: This work is based on the theoretical design of ref. 16 of the manuscript. However, in the experimental results section, the authors did not compare their results to the theoretical prediction of ref. 16. Can the authors make a comparison? What is the optimum sensitivity of this design? Why did the authors choose this wavelength range for the experimental demonstration? What is the optimum design?
Response 2: Thanks for your questions. Yes, this work is based on the theoretical design in ref. 16, which has shown a high RI sensitivity around 12550nm/RIU. However, ref. 16 is only a simulation paper. For LPG, the RI and temperature sensitivity could be reached to infinite, if we design the cladding mode resonance peak at the turning point. But for the experiment, it is difficult to fabricate the LPG working just at the turning point. Even in this manuscript, we have tried many times to inscribe the cascaded LPGs working around the turning point. So, for the sensing indicator, there is nothing comparable between these two papers. But the changing trend of sensitivity is consistent. And I also admit that the design of these two papers is the same, the only difference is that the Ref. 16 was from simulated results, and this works is from experimental results. We have taken several months to get these data.
For simulation, we could have the wavelength range as we wanted, but for experiment, we could only make our wavelength range close to the one we wanted. We wanted to have the same wavelength range as reported in Ref. 16, however, the fabrication process is too difficult to achieve that.
Point 3: p.2, line 52. “To verify this theory, we experimentally investigate in this work the transmission and sensing features of a cascaded LPG-based sensor operating near the PMTP.”The description is not clear, although the referee could get the point the author would like to say.
Response 3: Thank you for your suggestion. We have corrected that “in this work, we experimentally fabricate and investigate the transmission and sensing features of a cascaded LPG-based sensor operating near the PMTP” in line 55-56, which is highlighted by using “Tracking Changes” function.
Point 4: p.2, line 66. “As presented in [16], the resonance peak of LPG operated at PMTP would have an ultra-high sensitivity, due to the waveguide dispersion trend to the infinite.” Please be more specific. What does “trend to the infinite” mean?
Response 4: Thanks for your question. An LPG can couple the light from the fundamental core mode ( ) to a forward-propagating cladding mode ( ) at a resonance wavelength defined by the phase-matching condition , where is the grating period and is the core–cladding mode-index difference. Phase matching turning point corresponds to , with positive dispersion ( ) on the short wavelength side of PMTP, and the other side of PMTP has negative dispersion ( ). Obviously, as the slopes near the dispersion turning-point region are trending to the infinite, the coupled cladding modes near this region are, therefore, extremely sensitive to any external perturbation that may cause changes in grating period and mode index [17]. In Ref.16, the authors designed LPG working exactly at the PMTP, so it would have extra-high sensitivity.
[17] Xuewen Shu, Lin Zhang, Ian Bennion, Sensitivity characteristics near the dispersion turning points of long-period fiber gratings in B/Ge codoped fiber, Opt. Lett. 2001, 15, 1755-1757.
We have rewritten this sentence and made the full text logical. “LPG could couple the fiber core mode into the forward-propagating cladding mode, and in turn, it also could couple the cladding mode back to the core mode. Such reversible coupling behavior makes the LPG to construct an MZI. In this work, we experimentally fabricate a cascaded LPGs based MZI sensor.” was added in line 69-72, which is highlighted by using “Tracking Changes” function.
Point 5: p2, line 60-62. This paragraph is not needed.
Response 5: Thank you so much for pointing out this mistake, we have corrected this error. This paragraph was deleted.
Point 6: For the simulation result of Fig.2, why use 21st cladding mode, is there any specific reason?
Response 6: This is good question. The work presents that the cascaded LPGs working at the PMTP. According to our calculation, the turing point of 21st cladding mode is around C band. That is the reason we choose the 21st cladding mode. We have added the explanation in the manuscript.
“According to our calculation, the wavelength of turning point of the 21st cladding mode is with the C-band. By employing Eq. 5, we have simulated the temperature and RI sensitivity of the 21st cladding mode versus the wavelength around the turning point…” was added in line 96-99, which is highlighted by using “Tracking Changes” function.
Point 7: P.3, line99. What is a PS1250/1500 fiber?
Response 7: Thanks for your question. PS1250/1500 fiber is a photosensitive fiber which have the same fiber structure design as the standard SM28. Because of the fabrication method in our lab, we can only use such fiber to achieve LPG fabrication.
“which is a photosensitive single mode fiber” was added in line 110-111, which is highlighted by using “Tracking Changes” function.
Point 8: P.6, line 165. “These results also show that the bandwidth of the resonance peak of the cascaded LPG pair narrows with increasing temperature”How to see this from Fig. 7? How to see the bandwidth change? There is no changing temperature. Only temp sensitivity.From the results of Fig.7, it seems the temp sensitivity is about the same in the range between 1600 and 1750 nm. The authors should relate this to their findings above.
Response 8: Thanks for point out this. We have deleted this sentence. The peak bandwidth of cascaded LPG was also affected by the coupling strength of the grating.
Point 9: From the results in Table 4, it seems the sensitivity is highest when grating separation is 3 cm. It is not changing monotonically as the authors described in the conclusion (p.9, line 233). Can the author comment on why it’s highest at 3 cm? How many times did the authors repeat the measurement?
Response 9: This is good question and suggestion. Actually, we have discussed this in the manuscript. The period of LPG in this work is around 154 um. According to the simulation, such period LPG would have the turning point at the wavelength of 1570 nm. According to the simulated analysis, the closer the peak is to the turning point, the higher the sensitivity. “For the interference peaks on each side of the PMTP, the wavelength shift increases as the peak wavelength approaches 1570 nm (PMTP), so the sensitivity increases as the wavelength peak approaches the PMTP. Among the four samples, only the sample with the 3 cm separation between LPGs has a resonance peak located near the PMTP (around 1583 nm in air), so the above analysis indicates that this cascaded LPG pair would have the higher RI sensitivity. ” (line 215-220).

Reviewer 3 Report
The authos present a sensor based in two LPG near to turnin point in cascated. This is a Mach-Zehnder interferometer. The results no increase new ideas or system in the field. the interferometer in fiber and LPg in turning points are explored for several authors (same type the fibers and technique-UV).
The results are poor. The authors do not present a sensor or application only laboratory test .
i recommend reject this manuscript.
Author Response
We are afraid there are some misunderstandings of the reviewer. This manuscript has experimentally achieved extra-high sensitivity RI and temperature sensor with high Q-factor. The LPG and MZI based sensors were reported and published in many papers and the configuration of such sensor was theoretically proposed in Ref. 16, however, we believe that this manuscript is the first one to experimentally achieve the LPG based MZI working at the PMTP and also the discussion about the sensor sensing mechanism is given. We believe that any big progress in science is always the accumulation of small improvement. So, the context described in the manuscript is worth to be published in Sensors Journal.

Round 2
Reviewer 1 Report
I think that this revised manuscript has been corrected according to the reviewer's comments. Authors should carefully check the overall text, such as the typing error ( The cascaded LPGs construct an in fiber Mach-Zehnder interferometer )...
Author Response
Response: Thanks for the reviewer’s comments. We have carefully checked through the manuscript and corrected all the errors. All revised parts have been highlighted by red colour. We hope this manuscript is suitable for publishing in Sensors.

Reviewer 3 Report
The aauthors made some change in the manuscript, however, still don't show the advantage of this configuration. the authors not compare with other sensors.
The configuration of MZ is known. the sensitivity increase with length of cavity. this result are expect.
the results and sensor configuration no increase value in the field. I recommend reject
Author Response
Response: Thanks for the reviewer’s query. Every discussion always helps us to further improve the quality of manuscript. In fact, we agree that this paper is the experimental version of Ref. 16 (Ref. 16 is only a simulation work). And the idea is also sourced from Ref. 16. We don’t think that the experimental verify is worthless to be investigated, although we can calculate out the results. And sometime, the experiment results could be expected.
The reviewer said “the authors not compare with other sensors”, actually, in our original version, there was table 4 to compare the results with the other fiber sensors. According to the suggestion “For the manuscript simplification, I think that Table 1 and Table 4 are not necessary to be highlighted.” from the reviewer, we deleted this section, which we agreed that for the domain expert, they all know the performance.
The advantage of this configuration is that the cascaded LPGs operated at PMTP based MZI has achieved a relative high Q sensing performance with high sensitivity and measurement precision. In our simulation and experiment, we have found that the resonance wavelength closing to the turning point has the higher sensitivity, and the larger grating separation could also increase the sensitivity. We believe that the above information didn’t mention in the other paper. So, we think the manuscript is worth to be published, although there is only a small improvement.

Round 3
Reviewer 3 Report
The authors still not response to the previous questions.
the authors not study the influence os temperature and not clarify the efect os cavity length. Still not explain how discriminate the prositon of wavelength.
Not present clearly the advantege of this configuration.
In this moment is expect pratical applications in this area. the authors not propose one aplications.
I recomend reject.
Author Response
Response: Thanks for the reviewer’s comments. Our response is listed as following:
- “the authors did not study the influence of temperature”
We are sorry for the confusion of the temperature influence. Actually, the RI sensing experiment was operated inside an incubator under a temperature control system with 0.001℃ temperature control accuracy, which is better than the temperature measuring accuracy of the proposed sensor. So, we could ignore the temperature influence during the RI sensing experiment. We have added the description in the manuscript.
We add “The high temperature sensitivity of the proposed sensor would affect the RI measuring accuracy. To eliminate the temperature crosstalk, during the RI sensing process, the cascaded LPG sensors is placed inside incubator with 0.001℃ temperature control accuracy, which is better than the temperature measuring accuracy of the proposed sensor.” in line 176-179, which is highlighted by red colour.
- “not clarify the efect os cavity length”
We have added the discussion about the effect of the cavity length. According to the experiment and numerical analysis, the longer cavity length could improve the sensing performance in terms of sensitivity and precision. However, we can’t increase the cavity length unconditionally, the longer cavity could also increase the transmission loss.
“The transmission spectrum shows that the bandwidth of the resonance peaks of cascaded LPGs decreases linearly with increasing grating separation, and the Q factor increases simultaneously, in which the Q factors of cascaded LPGs with 2cm, 3cm, 4cm, and 5 cm separations are 35.8, 41.6, 53.3, 19.4, respectively. Compared with a conventional dual-peak LPG [26], the bandwidth and Q factor of cascaded LPGs with 5 cm separation are improved about four times. Although further increasing the separation gap should significantly increase the Q factor, the larger separation would increase the transmission loss of the cladding mode and reduce the interference peaks, thereby affecting the performance of the sensor.” and “The experiment results also indicate that the longer cavity length of cascaded LPG sensors has a higher sensitivity and better Q-factor to improve the sensing accuracy.” are added in line 126-133 and line 216-218, which is highlighted by red colour.
- “Still not explain how discriminate the prositon of wavelength”
The proposed sensor is a type of in-fiber interference. As the changing of temperature and SRI, the interference peak would change as well. For temperature sensing, we only need to record the peak position in the room temperature. And for RI sensing, we need to record the peak position in the water. And in the Fig. 4(b), there is not any additional or disappeared interference peak when the sensor is placed in air and water. So we can use the order of interference peak to discriminate the peak position.
- “Not present clearly the advantege of this configuration.”
LPG based sensors have a mode dependent sensitivity. By designing the LPG operated at the PMTP, the sensitivity could be greatly improved, however, the resonance peak at the PMTP has a broad peak which has a poor Q-factor. So, we have presented a cascaded LPG pairs working at the PMTP to improve the sensitivity and Q factor as well.
We have added the description in the manuscript.
“Benefit from the LPG working at near the PMTP, the proposed sensors have shown high sensitivity to both the surrounding refractive index (SRI) and temperature, and the cascaded structure would improve the Q factor of sensor.” is added in line 53-55, which is highlighted by red colour.
- “In this moment is expect pratical applications in this area. the authors not propose one aplications.”
Such high RI sensitivity of the proposed sensor would be suit for the biochemical sensing, which we have described in the manuscript. Currently, we are planning for the practical application in the biosensing, but this will be our next research work. In this work, we mainly focus on the characterization of the cascaded LPG operated at the PMTP based sensor.
We add “Such high RI sensitivity would promote the cascaded LPG-based sensor suiting for highly sensitive biochemical sensing applications.” in line 221-222, which is highlighted by red colour.
Additionally, we have made minor corrections in the manuscript which is highlighted by red colour.

Round 4
Reviewer 3 Report
the authors present and responde to previous questions. The present manuscript can be publish in present form.